# BINARY NODE CLUSTERING VIA CONTRASTIVE LEARNING FOR HAPLOTYPE PHASING IN DE NOVO GENOME ASSEMBLY

## ABSTRACT

Accurate haplotype phasing is essential for high-quality genome assembly, yet *de novo* phasing without parental data for complex genomes remains a challenge. We formulate phasing as a binary, overlapping node clustering problem on unitig graphs where nodes represent contiguous, nonbranching DNA sequence fragments and different edge types capture sequence overlaps as well as Hi-C proximity information. To solve this problem, we design a contrastive learning framework with custom objective functions and train a graph-transformer-based model termed grapHiC to distinguish nodes with paternal, maternal, or homozygous haplotypes. We show that grapHiC significantly outperforms other node clustering methods on genome-sized datasets and that grapHiC's predictions can successfully guide de novo genome assembly, producing well-phased assemblies across diverse human genome assembly graphs using the DipGN-Nome assembler. Our code, trained model, and dataset are available at `https://anonymous.4open.science/r/graphic_iclr-688D/` (repository anonymized for peer review).

## 1 INTRODUCTION

Genome assembly, referring to the reconstruction of the genomic sequence of an organism from a set of error-prone sequencing reads, is one of the most fundamental algorithmic challenges in computational biology. Recent advances have seen the integration of machine learning techniques into different steps of the genome assembly process (Schmitz et al., 2025; Vrček et al., 2025; Luo et al., 2024; Battistella et al., 2025; Xue et al., 2022; Ke & Vikalo, 2020a;b). Many organisms have multiple copies of their genome; these are referred to as haplotypes, and may differ from each other by containing variants inherited from one parent but not the other. Most animals, including humans, are diploid, possessing two copies of autosomal chromosomes.

Separating the different haplotypes of the genome – called *phasing* – is an important part of many genomics workflows such as population genomic analyses or functional studies, as the effects of a mutation on gene expression and epistatic interactions are often molecule-specific. Untangling the different haplotypes during genome assembly remains a major challenge, especially for complex genomes, even with recent rapid progress in genome sequencing and assembly techniques. The easiest way to separate haplotypes is using trio binning, which uses the genomes of an organism's parents to determine the origin of each variant. However, it requires extra lab work and is impractical for wild or undocumented samples where parents are unknown.

Recent work has demonstrated the potential of graph neural networks in *de novo* genome assembly. (Vrček et al., 2025) shows how to generate assemblies for the haploid case and (Schmitz et al., 2025) for diploid genomes using trio binning. These methods highlight the suitability of GNNs for making predictions on large and complex assembly graphs. The human genome provides an especially strong benchmark in this context: it is biologically challenging, and high-quality references are available to generate ground truth and enable proper evaluation. Previous deep-learning approaches to phasing such as (Battistella et al., 2025; Xue et al., 2022; Ke & Vikalo, 2020a;b) have focused on aligning genomic fragments to a reference genome, and then clustering them into haplotype partitions using a variety of approaches (see *Related Work* below). However, high-quality references are not available for many species and may inadequately represent the sample organism

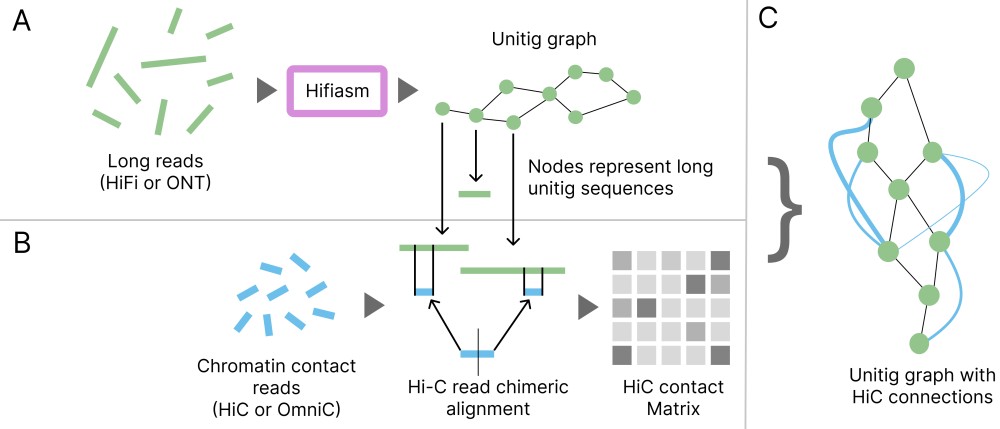

Figure 1: **A:** Long reads are processed by `hifiasm` to create a unitig graph. **B:** Chromatin Contact (Hi-C) reads are mapped against the unitigs, to create a contact matrix. **C:** The Contact matrix is realized as edges in the unitig graph, where the weight is defined by the amount of Hi-C contacts of unitigs.

in genetically diverse species. The alignment step can also introduce biases, especially in repetitive regions of the genome. All methods using alignment to a reference genome ultimately lead to some level of reference bias in downstream analyses, which can be hard to account for.

Chromatin Contact sequencing (Hi-C) measures the frequency of physical contacts between pairs of DNA loci, capturing their spatial proximity in the nucleus. These contacts can be used for haplotype phasing: loci on the same chromosome, and especially on the same haplotype, tend to interact more frequently than loci on different chromosomes or haplotypes. This long-range linkage information can therefore guide algorithms to assign variants to the correct haplotype. Hi-C sequencing does not require additional samples and has recently become affordable and routine, but is still subject to noise and a multitude of biases. Many recent phasing tools (Lorig-Roach et al., 2024; Ouchi et al., 2023; Zhang et al., 2024; Zhang, 2018; Cheng et al., 2021; Antipov et al., 2024) allow using Hi-C reads for phasing, and may optionally also support parental sequencing. These tools either make simplifying assumptions about the input assembly graph (such as being in bubble-chain format, or fully simplified) and then use simple statistical models or heuristics to resolve haplotypes, or require significant manual intervention by the user. They tend to work well on simple genomes with large amounts of Hi-C data, but have received little validation in more complex use cases. To our knowledge, there is currently no software using deep learning for reference-free phasing while incorporating Hi-C information. To fill this gap, we set out to design graph-transformer-based Hi-C phasing: `grapHiC`.

## 2 OVERVIEW

Our approach is based on unitig graphs – a compressed representation of sequencing reads. Nodes in this graph representation correspond to sequenced DNA fragments. An edge is added between two nodes if their DNA sequences overlap, i.e., if the suffix of one matches the prefix of the other. Sets of nodes forming an unbranching path, where each node has exactly one incoming and one outgoing edge within the set, are compressed into a single node, called unitig. By operating directly on the raw unitig graph, before any simplifications or reductions, we avoid errors that could arise from misjoining fragments belonging to different haplotypes. Unitig graphs can be very messy and large. To make them amenable to graph machine learning approaches, we choose a linear-time graph transformer architecture capable of integrating global information, and augment the graph with additional Hi-C information. Hi-C connections are added between nodes, with the edge weight indicating the strength of the Hi-C signal. Figure 1 illustrates how these graphs are created. On these heterographs, we formulate haplotype phasing as a binary node clustering problem with overlapping clusters: if the genomic sequence inherited from both parents is identical (homozygous) at a certain position,

a node can belong to both haplotypes. Without parental data, the haplotypes are interchangeable. For two variants on the same chromosome, it is possible to infer if they were inherited from the same or different parents, but not if that parent was the mother or the father. Any valid haplotype assignment thus remains valid after flipping the cluster labels of the maternal/paternal copies of any set of chromosomes. For two variants on different chromosomes, Hi-C-based connections cannot be used for phasing, as they are not on the same DNA molecule.

Due to these unusual invariants and the noise in the raw data, standard node classification approaches are insufficient. To address this, we define a custom objective function inspired by the pair loss commonly used in contrastive learning, designed to respect these symmetries. We call our loss 'supervised binary pair loss' (SBP-Loss). Figure 2 sketches the overall pipeline of training `grapHiC` and using it to phase a unitig graph.

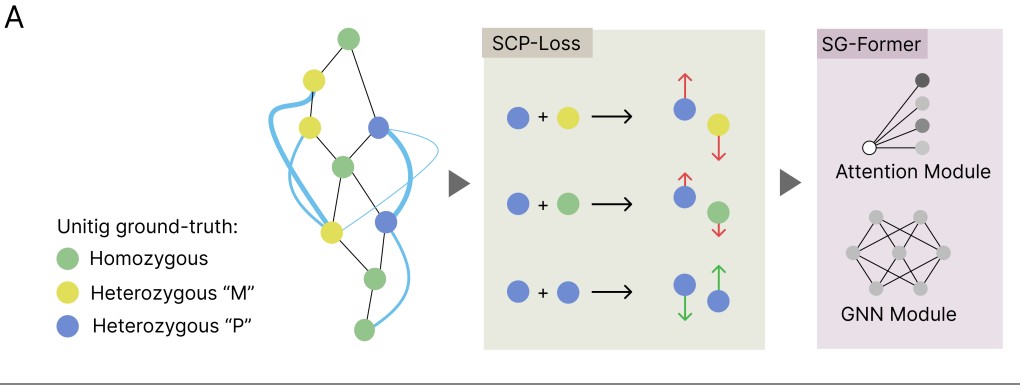

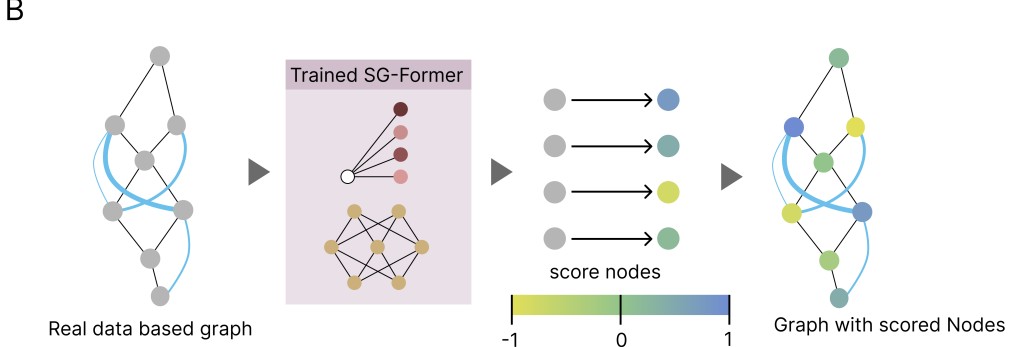

Figure 2: **A:** A training dataset of graphs is used to train an SG-Former (Wu et al., 2024) using SCP-Loss (proposed here) as training objective. **B:** The trained model gets an input graph, and outputs node scores, representing hapolotype allocations, and homozygousity probability.

In summary, our contributions are threefold: (1) we introduce the first machine-learning–based tool for reference-free phasing with Hi-C reads, and the first method capable of phasing raw unitig graphs; (2) we develop a data processing pipeline to construct Hi-C–augmented unitig graphs, enabling reproducible and automated data preparation; and (3) we define a novel objective function tailored to binary (fuzzy) clustering, showing that it outperforms standard contrastive losses and other baselines on the reference-free phasing task.

`grapHiC` is available at `https://anonymous.4open.science/r/graphic_iclr-688D/`(repository anonymized for peer review)

## 3 RELATED WORK

### 3.1 DEEP LEARNING FOR GENOME ASSEMBLY

In recent years, several approaches have been presented that leverage neural networks for various problems related to genome assembly. For *de novo* assembly, GNNome (Vrček et al., 2025) performs layout on an Overlap-Layout-Consensus (OLC) graph using a Graph Convolutional Network (GCN). However, GNNome assembles the multiple haplotypes present in the sampled reads into a single consensus haplotype. Šimunović et al. (Šimunović et al., 2025) focus similarly on the layout of De Bruijn graphs. DipGNNome (Schmitz et al., 2025) is the first machine learning-based *de novo* assembler to create phased assemblies, given parental data as additional inputs.

For reference-based haplotype phasing, several methods have been introduced. GAEseq (Ke & Vikalo, 2020b) and CAECseq (Ke & Vikalo, 2020a) were the first neural network-based frameworks. They used a graph auto-encoder and convolutional auto-encoder, respectively. NeurHap (Xue et al., 2022) demonstrated better performance by representing haplotype phasing as a graph colouring problem on read-overlap graphs obtained by aligning reads to a reference, and using a graph neural network for learning color assignments. However, these approaches have been evaluated on data sets that are much smaller than those encountered routinely in genomics. Instead of the $\approx 10^{5-7}$ reads required to assemble a real eukaryotic genome at high quality, only a few hundred reads were used to construct test and training overlap graphs. `ralphi`'s architecture is more scalable (Battistella et al., 2025). By using a deep reinforcement learning framework to optimize the maximum fragment cut (MFC), it is able to use a graph convolutional network not requiring all vs. all attention. This allows `ralphi` to scale to real-world datasets; however, it still uses a read overlap graph obtained by alignment to a reference genome. Reference-derived graphs are fundamentally different to *de novo* overlap/unitig graphs. They are much cleaner and contain far fewer spurious connections, making them more amenable to heuristics approaches (like MFC). Additionally, the MFC approach of `ralphi` cannot account for homozygous nodes or provide certainty estimates of the predicted haplotype assignments. This leaves the problem of phasing directly on reference-free raw unitig graphs at realistic scales open in the machine-learning-oriented literature. Overcoming the challenges associated with this setting promises the opportunity to phase genomes without requiring additional parental data or introducing reference bias.

### 3.2 CLUSTERING AND CONTRASTIVE LEARNING

Traditional node clustering approaches typically rely on the homophily assumption: the idea that nodes that are directly connected or located close together in a graph are more likely to belong to the same cluster.Spectral clustering (Ng et al., 2001) performs dimensionality reduction by using the eigenvectors of a graph Laplacian derived from the adjacency matrix, and then clusters the nodes in this reduced space based on their connectivity. The Louvain algorithm (Blondel et al., 2008) detects communities by maximizing modularity, identifying clusters with higher-than-expected internal connectivity. Label propagation methods (Zhu & Ghahramani, 2002) iteratively diffuse known labels across the graph under the assumption that adjacent nodes tend to share similar properties. In cases where clusters are defined not by direct connections but by structural roles or more complex interaction patterns, learned deep representations can offer a benefit. On these representations, simple clustering algorithms such as k-nearest neighbor (KNN) clustering (Peterson, 2009; Keller et al., 1985) can be used, enabling the detection of higher-order patterns beyond simple connectivity.

Representation learning approaches for graph clustering can fall into different categories such as reconstructive methods, often based on autoencoders (e.g. (Kipf & Welling, 2016)), (generative) adversarial methods (Mukherjee et al., 2019), and contrastive learning approaches (Watteau et al., 2024). Each of these strategies seeks to map nodes to a lower-dimensional embedding space where clustering is easier, but approaches this in different ways. Foundational work in establishing contrastive learning was published in (Chopra et al., 2005; Hadsell et al., 2006). Early graph-specific contrastive methods include the Linear Relational Encoding (LRE) strategy (Paccanaro & Hinton, 2000) and TransE (Bordes et al., 2013), applied to knowledge graph completion.
More recent advances in contrastive objectives, such as Supervised Contrastive Learning (SupCon) (Khosla et al., 2020) and InfoNCE (Oord et al., 2018), although developed outside the graph domain, illustrate how label information can be incorporated into contrastive frameworks.

## 4 SUPERVISED BINARY PAIR LOSS

We build upon the classic pair loss used in contrastive learning (Chopra et al., 2005; Hadsell et al., 2006), which aims to minimize distances between similar samples while enforcing a margin between dissimilar ones:

$$\mathcal{L}_{\text{pair}}(q, k) = \begin{cases} D^2(e_q, e_k) & \text{if } y_q = y_k \\ \max(0, m - D^2(e_q, e_k)) & \text{if } y_q \neq y_k \end{cases} \tag{1}$$

where $q$ and $k$ represent query and key samples, $e_q, e_k$ are the corresponding embeddings, $y_q, y_k$ are the class labels, $D^2$ is the squared Euclidean distance, and $m$ is a margin parameter.

During training, we have ground-truth labels for all nodes, distinguishing our haplotype phasing scenario from typical unsupervised clustering problems. Each node is assigned one of three labels: exclusively belonging to haplotype A ($y = -1$), exclusively belonging to haplotype B ($y = 1$), or homozygous (being shared by both haplotypes) ($y = 0$). These labels can be leveraged in our objective function during training.

Given the binary nature of our clustering task, we employ a one-dimensional embedding space where the model outputs scalar predictions $p \in [-1; 1]$ for each node. Our goal is to push nodes from different haplotypes toward opposite extremes ($-1$ and $1$) while centering homozygous nodes around $0$. For our one-dimensional case, the Euclidean distance reduces to the squared absolute difference $D^2(e_q, e_k) = (p_q - p_k)^2$.

Another key insight in our approach is that the margin should adapt to the label structure of our problem. We set the margin $m = |y_q - y_k|$, which naturally encodes the desired distances. When comparing nodes with identical labels ($y_q = y_k$), we have $m = 0$, pushing the predicted labels towards exact agreement. For a pair consisting of a homozygous node ($y = 0$) and a heterozygous node exclusive to one haplotype ($y = \pm 1$), we set $m = 1$. When comparing nodes from opposing haplotypes ($y = -1$ vs $y = 1$), we choose $m = 2$, requiring maximal separation.

By introducing an indicator variable $s_{qk}$ to capture the relationship between labels, we can derive a unified formulation:

$$s_{qk} = \begin{cases} -1 & \text{if } y_q = y_k \\ +1 & \text{if } y_q \neq y_k \end{cases} \tag{2}$$

This allows us to express our *Supervised Contrastive Pair (SBP) Loss* as:

$$\mathcal{L}_{\text{SBP}} = \frac{1}{|\mathcal{B}|} \sum_{(q,k) \in \mathcal{B}} \max(0, |y_q - y_k| - s_{qk} \cdot (p_q - p_k)^2) \tag{3}$$

where $\mathcal{B}$ is a batch of node pairs sampled from the graph. By varying how we sample the node pair batches, we can make a model learn with focus on different aspects. We define a *local* and *global* version of the SBP loss corresponding intuitively to two commonly used error metrics in genome phasing, namely the Hamming and switch error.

**Global SBP loss:** For each node $q$ in the graph, we randomly select another node $k$ to form a pair $(q, k)$ for the contrastive loss computation. To do this efficiently in parallel, we convert the full set of nodes in the graph into a list and apply a random permutation, ensuring that each node appears exactly once as a query node, paired with a randomly selected key node.

**Local SBP loss:** For each node $q$, we randomly sample a node $k$ sharing an overlap edge with $q$. These represent sequences that are located nearby along the chromosome and incentivize locally coherent predictions.

The SBP loss exhibits distinct behavior depending on the label relationship. When both nodes have the same label ($y_q = y_k$), the loss becomes $(p_q - p_k)^2$, directly penalizing any difference with no margin tolerance. When nodes have different labels ($y_q \neq y_k$), the loss encourages predictions to

be separated by at least $|y_q - y_k|$, with violations penalized quadratically. This formulation naturally guides the model toward the desired clustering structure while leveraging all label information available in our supervised problem setting. In analogy to the pair losses, we also create a supervised binary triplet (SBT) loss (detailed in Appendix A). We use an additional auxiliary loss to push predictions of nodes shared between clusters towards zero using a simple mean squared error. This can help the model converge earlier during training:

$$\mathcal{L}_{\text{aux}} = \frac{1}{|N_0|} \sum_{k \in N_0} p_k^2 \tag{4}$$

where $N_0 := \{k | y_k = 0\}$, and $p_k$ are the corresponding predictions.

## 5 MODEL ARCHITECTURE

Given an input graph with adjacency matrix $\mathbf{A}$ and node feature matrix $\mathbf{X} \in \mathbb{R}^{n \times d}$, the features are first embedded into an initial node embedding $\mathbf{Z}^{(0)}$ via a feed-forward input layer $f_{\text{in}}$:

$$\mathbf{Z}^{(0)} = f_{\text{in}}(\mathbf{X}) = \mathbf{X}\mathbf{W}_{\text{in}} + \mathbf{b}_{\text{in}} \tag{5}$$

`grapHiC`'s model architecture is based on the simplified graph transformer (SG-Former) (Wu et al., 2024), consisting of a GNN module and a linear attention (LA) module. The GNN captures local graph topology, while the attention models global interactions. The final representation is:

$$\mathbf{Z}_O = (1 - \alpha) \, LA(\mathbf{Z}^{(0)}) + \alpha \, \text{GNN}(\mathbf{Z}^{(0)}, \mathbf{A}) \tag{6}$$

$$\hat{y} = \tanh(f_{\text{out}}(\mathbf{Z}_O)) \tag{7}$$

where $\alpha$ is a hyperparameter balancing the GNN and attention representations, $LA(\mathbf{Z}^{(0)})$ is the linear attention module, and $f_{\text{out}}$ is a final feed-forward layer.

The GNN is implemented as a heterogeneous GIN (Xu et al., 2018) to handle multiple edge types $t \in \{o, h\}$ (overlap and Hi-C) with edge-separated adjacency matrices $\mathbf{A}_t$ and weights $\mathbf{W}_t$. A single heteroGIN layer updates node embeddings as:

$$\mathbf{Z}^{(k+1)} = \sum_{t \in \{o, h\}} \text{GINConv}_t(\mathbf{Z}^{(k)}, \mathbf{A}^{(t)}, \mathbf{W}^{(t)}) \tag{8}$$

where each $\text{GINConv}_t$ computes:

$$\text{GINConv}_t(\mathbf{Z}^{(k)}, \mathbf{A}^{(t)}, \mathbf{W}^{(t)}) = \text{MLP}_t \left( (1 + \epsilon_t)\mathbf{Z}^{(k)} + \sum_{(i,j) \in \mathbf{E}^t} \mathbf{W}_{ij}^{(t)} \, \mathbf{Z}_j^{(k)} \right) \tag{9}$$

with learnable scalar $\epsilon_t$ and edge weight $\mathbf{W}_{ij}^{(t)}$. The full GNN module $\text{GNN}(\mathbf{Z}^{(0)})$ denotes a stack of $l$ such heteroGIN layers with residual connections and normalization.

This design allows the model to capture relation-specific patterns while properly weighting Hi-C edges by contact strength. Multiple heteroGIN layers are combined with global-attention-based embeddings in the prediction head, preserving both local graph topology and global interactions.

## 6 HI-C ENHANCED UNITIG GRAPHS

### 6.1 DATA PIPELINE

We construct Hi-C–enhanced unitig graphs from a set of simulated HiFi and real Hi-C reads from the same genome. HiFi reads are sampled from the haplotype-resolved I002C genome sequence using `PBSim3`(Sarashetti et al., 2024b). For the Hi-C reads, we use Omni-C reads used in the original genome assembly. `PBSim3` aims to emulate the error profile observed in PacBio Hi-Fi-based genome sequencing; using simulated data allows us to have ground truth position and haplotype information for each read , which we can use as labels for training and validation. To increase diversity across training graphs, we use different replicates of the Hi-C experiments and randomly

subsampled HiFi read sets to construct each graph. Unitig graphs are generated from HiFi reads using `hifiasm` v0.25 (Figure 1 **A**), then the Hi-C reads are mapped against the unitigs (Figure 1 **B**).

For this mapping step, we implement two alternative pipelines: an **accurate** pipeline and a **fast** pipeline. The accurate pipeline is based on the nf-core/hic pipeline and uses base-level alignment with `bowtie2` (Langmead & Salzberg, 2012) followed by normalization using `Cooler` (Abdennur & Mirny, 2019). The fast pipeline replaces exact alignment with a minimizer-based mapping using `minimap2` (Cheng et al., 2021) and applies symmetric adjacency normalization. See Appendix B for detailed descriptions of both pipelines. The output of the data preparation pipeline is an augmented unitig graph containing both sequence overlaps and Hi-C contacts between unitigs as edges. Hi-C edges have a weight indicating the strength of the connection between the nodes (Figure 1 **C**). Nodes in the graph have several features: the number of adjacent overlaps and Hi-C edges, the length of the sequence of the unitig, and the amount of reads supporting the unitig as reported by `hifiasm`. Details of the feature extraction are given in Appendix C.

## 6.2 DATASET

We train our model on two synthetic datasets consisting of synthetic reads sampled to 40x coverage from the I002C phased human genome (Sarashetti et al., 2024b): **Dataset (I)** is a single chromosome dataset created with the **accurate** pipeline consisting of 45 chromosome-level graphs. **Dataset (II)** is a dataset created with the **fast** pipeline consisting of 35 full-genome graphs. More details about the dataset are given in Appendix D. Training details of `grapHiC` can be viewed in Appendix E, and the resources used for training are available in Appendix F.

## 7 EVALUATION: CLUSTERING

To evaluate the clustering capability, we compare accuracy (of the best one-to-one mapping between predicted cluster labels and true labels, using the Hungarian method), Adjusted Rand Index (ARI) and Normalized Mutual Information (NMI)(Hubert & Arabie, 1985; McDaid et al., 2013). Since these methods can only evaluate disjoint clustering, we also include the Omega Index (Collins & Dent, 1988), an extension of the ARI for non-disjoint (fuzzy) solutions. For evaluation, we show `grapHiC`'s performance on the four datasets compared to three traditional clustering algorithms. We choose spectral clustering (Ng et al., 2001), Louvain algorithm (Blondel et al., 2008) , and Label Propagation (Zhu & Ghahramani, 2002) to cluster nodes into haplotypes.

We compare different versions of `grapHiC` trained all trained with **Dataset (I)** but with different objective functions and evaluate their performance against our proposed SBP-Loss. We include both SBP-Loss (global) and SBP-Loss (local), as well as the SBT-Loss described in Section 4. We compare our losses to a supervised version of InfoNCE (Oord et al., 2018) and the Supervised Contrastive Loss (SupCon) (Khosla et al., 2020), each with a 64-dim embedding output and an additional 16-dim projection head that is not used during evaluation. Similarly to our proposed losses, both of these approaches use batchwise processing of 16 negative (and in the case of SupCon also 16 positive) samples. The clustering is performed using $k$-means clustering for Accuracy, NMI, and ARI metric computation and Fuzzy $c$-Means clustering (Bezdek et al., 1984) for the Omega Index computation. Table H shows the Accuracy and Omega Index of the compared methods. Appendix G shows the results graphically and adds NMI and ARI results.

We conduct an ablation study to assess the impact of each component of `grapHiC` on prediction quality. Overlap and HiC-edges, as well as the GNN component, are absolutely required by `grapHiC`. Removing them makes the predictions essentially random. Removing the attention component or the auxiliary loss mildly decreases performance. The full results are given in Appendix H.

## 8 EVALUATION: GENOME ASSEMBLY

To evaluate our model in a real-world use case, we test whether `grapHiC` predictions can be useful for *de novo* genome assembly. For this, we use the recently published assembler `DipGNNome` (Schmitz et al., 2025) and a `grapHiC` model trained on **Dataset (II)**. We use `grapHiC` to assign

| Method | chr10 | | chr19 | | chr15 | | chr22 | |
|---|---|---|---|---|---|---|---|---|
| | Acc | Omega | Acc | Omega | Acc | Omega | Acc | Omega |
| Spectral | 0.503 | 0.002 | 0.515 | -0.172 | 0.514 | -0.069 | 0.513 | -0.077 |
| Louvain | 0.513 | 0.000 | 0.510 | -0.009 | 0.511 | -0.001 | 0.519 | -0.002 |
| LP | 0.507 | -0.000 | 0.520 | -0.061 | 0.511 | -0.001 | 0.505 | -0.015 |
| SBP Loss (global) | 0.876 | **0.548** | **0.871** | **0.530** | 0.852 | **0.466** | **0.860** | **0.501** |
| SBT Loss | 0.871 | 0.527 | 0.850 | 0.535 | **0.855** | 0.476 | 0.831 | 0.441 |
| SBP Loss (Local) | 0.855 | 0.470 | 0.853 | 0.316 | 0.840 | 0.391 | 0.825 | 0.368 |
| SupCon | 0.653 | 0.491 | 0.767 | 0.210 | 0.659 | 0.131 | 0.843 | 0.143 |
| InfoNCE | **0.880** | 0.496 | 0.831 | 0.266 | 0.844 | 0.414 | 0.855 | 0.366 |

Table 1: Comparison of different phasing methods on the chromosome validation sets. Accuracy and Omega Index reported are averaged for all replicates per chromosome. Best scores are marked in bold.

a haplotype score to each node in the graph. These scores are then used in the beam search process of `DipGNNome` to separate the haplotypes during the assembly process. Appendix I provides more details on how `grapHiC`'s predictions are used to find assemblies using `DipGNNome`. We evaluate the combined `D-grapHiC` (`DipGNNome` + `grapHiC`) pipeline on synthetic full-genome samples generated by `PBsim3` (Ono et al., 2022), with read error profiles and length distributions matching real HG002 HiFi reads. The resulting assemblies are compared against those produced by `hifiasm (hic)`. For evaluation, we use synthetic human unitig graphs supplemented with Hi-C reads, randomly downsampled to 50 million pairs for performance reasons.

**Genome Assembly Experiments**  We evaluate `D-grapHiC` on two full-genome synthetic graph settings: (i) I002C, the genome used for training (but graphs created out of reads not seen during training), and (ii) HG002, a different high-quality human genome. Graphs generated from different read sets of the same genome can exhibit substantially different topologies and attributes. We therefore assess how performance varies across diverse graph topologies both within a single genome and across different human genomes.

Table 2 shows that the performance of `D-grapHiC` on the synthetic I002C and HG002 assembly graphs is overall comparable to that of `hifiasm`. Phasing accuracy, measured by switch error, is nearly identical: `hifiasm` exhibits slightly lower error for three haplotypes, whereas `DipGNNome` shows slightly lower error for eight haplotypes. Regarding Hamming error, `DipGNNome` is moderately higher than `hifiasm` across all haplotypes except HG002(3) haplotype 1. However, the original `DipGNNome` paper also noted higher Hamming error rates than `hifiasm`, suggesting that the difference may not be attributable to `grapHiC`'s predictions.

The `grapHiC`-guided `DipGNNome` assemblies achieve comparable but usually lower NG50 contiguity than `hifiasm`, with an exception at I002C(1), where NG50 is higher for haplotype 1. Duplication rates are consistently lower for `DipGNNome`. Altogether, this shows that `grapHiC`-guided assemblies produced by `DipGNNome` can produce assemblies close to state-of-the-art in terms of both contiguity and phasing quality while, in some cases, avoiding erroneous duplications.

## 9 CONCLUSION

We present `grapHiC`, the first machine-learning–based framework to phase reads without a reference assembly. It is also the first tool to perform phasing on an unsimplified unitig graph. `grapHiC` uses Hi-C as well as overlap information, and can distinguish homozygous from heterozygous regions. To develop `grapHiC`, we introduce the Supervised Binary Pair Loss, an objective function for binary (fuzzy) clustering with a batch mode tailored for node clustering on graphs with ground-truth labels. This loss accounts for the inherent label-switching symmetry and thus outperforms alternative contrastive objectives in the phasing problem, improving both performance and training

Table 2: Comparison of `D-grapHiC` and `hifiasm` on different I002C and HG002 replicates. Values are shown as P/M (paternal/maternal) in Megabasepairs (Mb). NG50 shows the performance on each haplotype. Hamming and switch error rates were computed using `yak` (Cheng et al., 2021), while other metrics were computed with `MiniGraph` (Cheng et al., 2021). Bold indicates the best value for each haplotype.

| Dataset | Metric | Hifiasm | D-grapHiC |
|---|---|---|---|
| **I002C(1)** | Length (Mb) | 5561.3 / 5561.3 | 2626.2 / 2619.4 |
| | Rdup (%) | 0.5 / 0.5 | **0.04 / 0.00** |
| | NG50 (Mb) | 82.2 / **82.2** | **86.8** / 66.8 |
| | Switch Err (%) | 0.40 / 0.46 | **0.28 / 0.45** |
| | Hamming Err (%) | **0.72 / 0.81** | 1.58 / 2.26 |
| **I002C(2)** | Length (Mb) | 2785.3 / 2769.1 | 2626.7 / 2622.6 |
| | Rdup (%) | 0.53 / 0.51 | **0.03 / 0.01** |
| | NG50 (Mb) | **86.1 / 85.9** | 66.8 / 69.9 |
| | Switch Err (%) | 0.42 / 0.46 | **0.28 / 0.45** |
| | Hamming Err (%) | **0.87 / 0.99** | 1.02 / 1.82 |
| **I002C(3)** | Length (Mb) | 2764.8 / 2800.3 | 2633.9 / 2618.4 |
| | Rdup (%) | 0.43 / 0.71 | **0.01 / 0.03** |
| | NG50 (Mb) | **86.1 / 92.1** | 83.9 / 78.6 |
| | Switch Err (%) | 0.43 / **0.45** | **0.27** / **0.45** |
| | Hamming Err (%) | **0.97 / 0.87** | 1.23 / 1.48 |
| **HG002(1)** | Length (Mb) | 2881.2 / 2854.0 | 2690.7 / 2674.4 |
| | Rdup (%) | 0.73 / 0.47 | **0.05 / 0.01** |
| | NG50 (Mb) | **90.0 / 89.6** | 68.6 / 73.2 |
| | Switch Err (%) | **1.39** / 1.45 | 1.74 / **1.11** |
| | Hamming Err (%) | **2.09 / 1.93** | 3.11 / 2.48 |
| **HG002(2)** | Length (Mb) | 2868.8 / 2866.1 | 2679.2 / 2677.0 |
| | Rdup (%) | 0.64 / 0.84 | **0.03 / 0.01** |
| | NG50 (Mb) | **89.0 / 86.5** | 72.2 / 46.3 |
| | Switch Err (%) | **1.51** / 1.33 | 1.75 / **1.13** |
| | Hamming Err (%) | **1.92 / 1.64** | 3.15 / 2.93 |
| **HG002(3)** | Length (Mb) | 2872.5 / 2866.5 | 2669.6 / 2679.5 |
| | Rdup (%) | 0.68 / 0.60 | **0.04 / 0.01** |
| | NG50 (Mb) | **89.8 / 86.7** | 56.4 / 67.2 |
| | Switch Err (%) | **1.41** / 1.44 | 1.72 / **1.15** |
| | Hamming Err (%) | 3.03 / **2.54** | **2.92** / 3.19 |

stability. An auxiliary loss is used to center predictions for homozygous nodes and achieve faster convergence.

In experiments on synthetic datasets, `grapHiC` consistently outperformed traditional clustering methods. Applied to multiple human datasets, it proved effective for guiding genome assembly: when combined with `DipGNNome`, `grapHiC` produced well-phased assemblies with switch and Hamming error rates between 0–3%. The integration of `grapHiC` information into `DipGNNome` is broadly applicable and resembles an internal step in the `hifiasm` Hi-C mode, allowing possible integration into different assemblers. More generally, our results show that deep-learning–based haplotype prediction can integrate noisy Hi-C information, and provide valuable guidance in diploid genome assembly.

Future work could include benchmarking the method on non-human real-world datasets. An especially promising direction is the extension to polyploid genomes, where disentangling multiple haplotypes poses an even greater challenge largely unaddressed by all current phasing approaches. Such advances would move toward a universal, learning-driven framework for genome assembly.

**Reproducibility statement**  The data used to create the training and evaluation dataset is publicly available, as described in Appendix J. Our self-created resources such as code, trained model, and training dataset are available as described in Appendix K.

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

## A  SUPERVISED BINARY TRIPLET LOSS

**SBT Loss**  We create a supervised contrastive triplet (SBT) loss, based on the triplet loss common in contrastive learning. For this, we consider triplets of samples $(q, k_p, k_n)$ where $q$ is the query/anchor sample, $k_p$ is a positive key sample (same class as query), and $k_n$ is a negative key sample (different class from query). The general triplet loss can be formulated as:

$$\mathcal{L}_{\text{triplet}}(q, k_p, k_n) = \max(0, D(e_q, e_{k_p}) - D(e_q, e_{k_n}) + m) \tag{10}$$

Where $D$ is a distance function, $e_q$, $e_{k_p}$, and $e_{k_n}$ are the embeddings of the query, positive key, and negative key samples respectively, and $m$ is a margin parameter. In our binary haplotype phasing context, using one-dimensional embeddings, and batches over all nodes in the graph, this becomes:

$$\mathcal{L}_{\text{SBT}} = \frac{1}{N} \sum_{(q, k_p, k_n) \in \mathcal{B}} \max(0, |p_q - p_{k_p}| - |p_q - p_{k_n}| + m) \tag{11}$$

Where $p_q$, $p_{k_p}$, and $p_{k_n}$ are the scalar predictions for the query, positive key, and negative key samples. For finding query samples, we select random nodes from the set of desired classes, where nodes of shared clusters are added to the positive set. (Note that this takes longer than the selection of samples in pair loss (global), where we only need to shuffle the list of nodes once and use it for the whole batch). The margin $m$ is set to 1.

$$\phi_r(h_u, e_{uv}) = \text{MLP}_r(\text{concat}(h_u, e_{uv})) \tag{12}$$

### A.1  EDGE EMBEDDING UPDATES

Edge embeddings are updated through bidirectional message passing:

$$e_{uv}^{(l+1)} = e_{uv}^{(l)} + \psi^{(l)}(h_u^{(l)}, h_v^{(l)}, e_{uv}^{(l)}) \tag{13}$$

$$\psi^{(l)}(h_u, h_v, e_{uv}) = \text{MLP}_{\text{edge}}^{(l)}(\text{concat}(h_u, h_v, e_{uv})) \tag{14}$$

where $\psi^{(l)}$ is a learnable edge update function that incorporates information from both endpoint nodes and the current edge state.

This heterogeneous formulation allows the model to learn distinct interaction patterns for structural (overlap) and functional (Hi-C contact) relationships while maintaining computational efficiency through type-specific parameter sharing.

## B  DETAILS: HI-C ALIGNMENT PIPELINES

### B.1  ACCURATE HI-C ALIGNMENT PIPELINE

To accurately align Hi-C reads to the unitig graph, we use the `nf-core/hic` pipeline (Servant et al., 2023), which is based on `HiC-Pro` (Servant et al., 2015). A high-level overview of the dataset preparation procedure is shown in Suppl. Figure 3.

The pipeline first performs read quality control with `FastQC` and `MultiQC` (Ewels et al., 2016), then aligns the reads to the unitigs using `bowtie2` (Langmead & Salzberg, 2012). Unlike the standard practice in Hi-C analysis for chromosomal contact discovery, we use sensitive local alignment, since the unitigs may still contain sequencing errors that are only corrected in the final assembly stage. A custom script employing `Cooler` (Abdennur & Mirny, 2019) aggregates Hi-C connections per unitig.

### B.2  APPROXIMATE HI-C MAPPING PIPELINE

The approximate mapping pipeline replaces base-level alignment with a minimizer-based approach. Specifically, Hi-C reads are mapped to unitigs using `minimap2` instead of `bowtie2`, and ICE normalization via `Cooler` is replaced with symmetric degree normalization of the adjacency matrix .

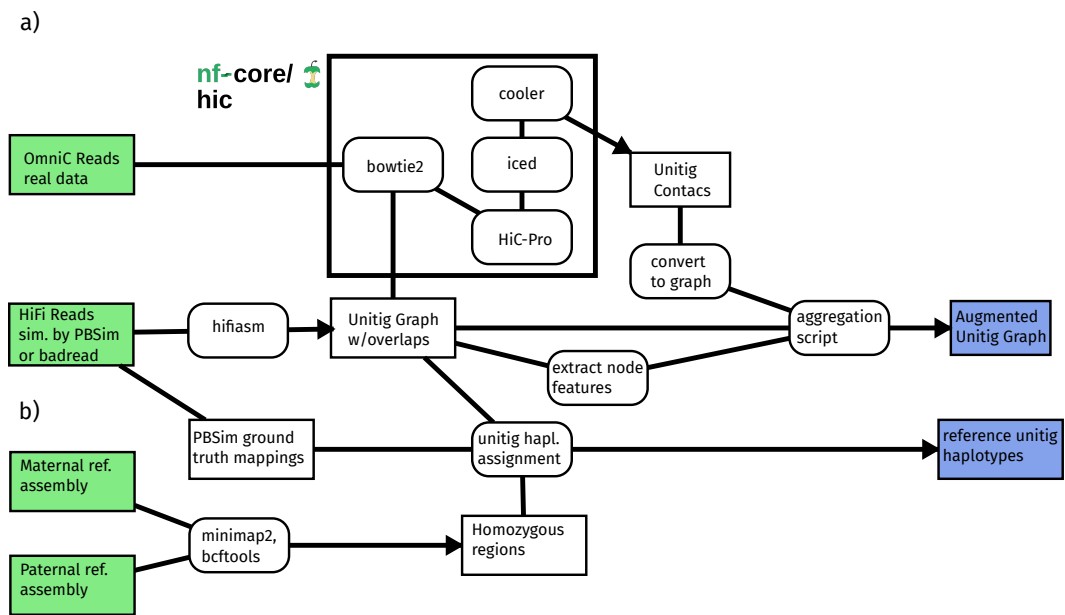

Figure 3: Pipeline used for dataset construction. Tools are shown in boxes with rounded edges, while intermediate outputs are shown in rectangular boxes. Initial inputs are highlighted in light green; outputs used for model training and evaluation are shown in light blue. **a)** To construct the augmented unitig graph, we generate unitigs with `hifiasm` on reads simulated with `PBSim` (PacBio HiFi) or `badread` (ONT). OmniC reads are mapped to the unitigs using the `nf-core/hic` pipeline based on the `HiC-Pro` toolchain. Contacts obtained from this process are added as additional edges to the existing overlap graph, with weights derived from ICE-normalized Hi-C contact frequencies. **b)** To obtain ground-truth haplotype information for phasing loss calculation, we combine the read simulator's origin loci for all reads within each unitig. Additionally, homozygous regions are identified by variant calling between the assembled parental genomes and subsequently annotated.

The resulting adjacency matrix $A$ is further symmetrically degree-normalized:

$$A_{\mathrm{norm}} = D^{-\frac{1}{2}} A D^{-\frac{1}{2}},$$

where $D$ is the diagonal degree matrix with entries $D_{ii} = \sum_j A_{ij}$. Below is the detailed command used to map Hi-C reads against unitigs:

```
# Mapping Hi-C reads against unitigs using minimap2
minimap2 -x sr -N 10 -p 0.8 -t <threads> \
    <unitigs.fasta> <hic_R1.fastq.gz> <hic_R2.fastq.gz> \
    > alignments.paf
```

Here, `-x sr` configures `minimap2` for short-read mapping (appropriate for Hi-C data), `-N 10` outputs up to ten secondary alignments per read, and `-p 0.8` specifies the minimum identity threshold. We then parse the PAF file and generate a weighted contact edge list with a custom script. Contact edges are then weighted by the number of supporting read pairs.

## B.3 DATA PIPELINE OUTPUT GRAPHS

The output of the data preparation pipeline is a heterograph where nodes represent unitigs and edges represent either sequence overlaps or Hi-C contacts. Each node is annotated with features extracted during preprocessing, and edges are labeled by type and assigned weights.

The full graph is serialized into the `.pyg` format using `networkx` and custom scripts. This makes it directly consumable by PyTorch Geometric-based models.

## C  NODE AND EDGE FEATURES

**Node features.** Each unitig node has four primary features:

- **Sequence length:** Computed from the FASTA sequences of the unitigs and divided by 10,000 to scale values into a range typically within $[0, 10]$.
- **Coverage:** Read coverage is extracted directly from the unitig metadata in the GFA output of hifiasm. Specifically, each `S`-line includes the number of reads supporting the unitig, which we use as an unnormalized scalar feature.
- **Degree:** The degree or number of adjacent Hi-C and overlap edges are each recorded as separate features, capturing how connected each unitig is in the graph under each edge type.

**Edge features.** Edges are annotated with both a type label (either `overlap` or `hic`) and a weight:

- **Overlap edges:** These represent exact or near-exact overlaps between unitigs as derived from hifiasm. All overlap edges are assigned a constant weight of 1.
- **Hi-C edges:** These reflect long-range physical proximity contacts inferred from Hi-C sequencing. The weight is the ICE and equation **??** normalized contact entry between the two connected unitigs.

## D  DATASET DETAILS

### D.1  DATASET (I): SINGLE CHROMOSOME

We evaluate our model on three synthetic datasets consisting of synthetic reads sampled to 40x coverage from the I002C phased human genome (Sarashetti et al., 2024b).

Each dataset includes all autosomes except Chromosome 14. We choose two non-acrocentric chromosomes as validation datasets ($2 \times 3$ graphs) and two acrocentric as well as two non-acrocentric chromosomes as test datasets ($4 \times 3$ graphs), and the rest for training ($16 \times 3$ graphs, containing the other two acrocentrics). Acrocentric chromosomes contain the long and complex rDNA repeats encoding the ribosomal RNAs that are challenging to assemble and phase, and thus represent a particularly tough challenge for our model. The four graphs in the test set are Chromosome 10 (medium size, non-acrocentric), Chromosome 15 (medium size, acrocentric), Chromosome 19 (short, non-acrocentric) and Chromosome 22 (short, acrocentric).

The training dataset consists of 45 graphs with a total of 139,574 nodes (mean: 3,101.64, std: 1,144.56) ranging from 982 to 5,427 nodes in each graph, where each unitig corresponds to tens to hundreds of thousands of base pairs of DNA. The graphs contain in total 1,043,808 (mean: 23,195.73, std: 31,544.18) sequence overlap edges, with a range of 1,355 to 107,087 in each graph and in total 2,584,249 (mean: 57,427.76, std: 28,061.06) HiC-connections with a range of 9,314 to 101,487 in each graph.

### D.1.1  DATASET (II): FULL GENOME

For training the new `grapHiC` model, we use and augment the dataset presented in `DipGNNome` (Schmitz et al., 2025), consisting of 35 graphs based on the I002C (human) genome. To introduce variation, we randomly subsample 50 million Hi-C read pairs from the original dataset for each graph. The resulting graphs combine real Hi-C data with synthetic HiFi data generated using `PBSIM3`. We split the dataset into 30 graphs for training and 5 for validation.

## E  TRAINING SETUP

We train grapHIC using a training pipeline implemented in PyTorch, with PyTorch-Geometric.

Table 3 summarizes the key training parameters.

For optimization, we use the Adam optimizer with a learning rate of $10^{-4}$. The training process supports multiple loss functions, including global pairwise loss, local pairwise loss, and triplet loss

Table 3: SGFormer Training Parameters

| Parameter | Value |
|---|---|
| GCN Layers | 8 |
| Transformer Layers | 3 |
| Hidden Dimension | 256 |
| Learning Rate | $10^{-4}$ |
| Auxiliary Loss Weight | 0.02 |
| Layer Normalization | True |
| GNN Dropout | 0.1 |

for prediction-based approaches, as well as contrastive embedding losses (SupCon and InfoNCE) for embedding-based approaches. An auxiliary loss with a weight of 0.02 is applied to encourage shared embeddings to approach zero.

During training, we evaluate model performance using the validation loss. We let the models run for 2000 epochs and select the one with the lowest loss on the validation dataset (see 6.2). The training configuration includes dropout regularization and layer normalization in each GNN layer block. We use Weights & Biases (Biewald, 2020) for experiment tracking, logging metrics such as loss values and accuracies.

## F   COMPUTE RESOURCES

All runs of the dataset preparation pipeline were performed on two nodes with 2 TB of RAM and 64 cores in 3-5 days. The pipeline was configured so each run would use at most 512 GB of RAM and 16 cores by setting Nextflow's job limits to the appropriate value. These steps, along with the conversion of the hifiasm output to a graph representation, also account for the major performance bottlenecks in the pipeline. The latter step involves a lot of memory operations in a Python script and presents an opportunity for runtime improvements even after considerable optimization of the implementation. In particular, meaningful parallelization could speed this task up immensely.

Because the pipeline is implemented in Nextflow, it natively supports most common compute infrastructures like local execution, AWS, Google Cloud, and a variety of HPC scheduling systems. The nf-core parts of the pipeline support all container architectures used by the nf-core project (docker, singularity, and conda environments). The aggregation steps at the end of the pipeline require a conda environment containing Cooler(Abdennur & Mirny, 2019) and NetworkX(Hagberg et al., 2008).

Execution of the pipeline requires considerable scratch space (up to a few hundred GB), as temporary read alignment files are written to disk multiple times if the Hi-C read alignment is broken up into chunks to reduce memory usage.

## G   CLUSTERING RESULTS DETAILS

This section shows the clustering experiment results in more detail. Suppl. Figure shows the comparison of `grapHiC` against classical clustering methods graphically. Table shows additionally ARI and NMI metrics for the comparison of different contrastive learning losses to train `grapHiC`.

## H   ABLATION STUDY

Suppl. Figure 5 shows our ablation study. We can show that each component is necessary to get best results with `grapHiC`.

| Method | chr10 | | chr19 | | chr15 | | chr22 | |
|---|---|---|---|---|---|---|---|---|
| | NMI | ARI | NMI | ARI | NMI | ARI | NMI | ARI |
| Spectral | 0.000 | 0.000 | 0.003 | 0.000 | 0.003 | 0.000 | 0.002 | 0.000 |
| Louvain | 0.001 | 0.000 | 0.001 | 0.000 | 0.000 | 0.000 | 0.001 | 0.000 |
| LP | 0.000 | 0.000 | 0.002 | 0.001 | 0.000 | 0.000 | 0.000 | -0.001 |
| SCP Loss (global) | 0.469 | 0.567 | **0.451** | **0.551** | **0.407** | 0.498 | **0.438** | **0.521** |
| SBT Loss | 0.448 | 0.549 | 0.411 | 0.501 | 0.409 | **0.506** | 0.378 | 0.439 |
| SCP Loss (Local) | 0.414 | 0.505 | 0.410 | 0.503 | 0.377 | 0.463 | 0.369 | 0.425 |
| SupCon | 0.185 | 0.197 | 0.321 | 0.383 | 0.172 | 0.191 | 0.433 | 0.473 |
| InfoNCE | **0.479** | **0.577** | 0.383 | 0.451 | 0.396 | 0.478 | 0.406 | 0.504 |

Table 4: Comparison of clustering methods by chromosome using NMI and ARI metrics. Best scores in each column are highlighted in bold. SCP Loss (Global) and InfoNCE demonstrate superior performance across chromosomes.

| Method | chr10 | | chr19 | | chr15 | | chr22 | |
|---|---|---|---|---|---|---|---|---|
| | Acc | Omega | Acc | Omega | Acc | Omega | Acc | Omega |
| Baseline | 0.876 | 0.548 | 0.871 | **0.530** | 0.852 | 0.466 | 0.860 | 0.501 |
| No Graph | 0.503 | -0.048 | 0.509 | -0.132 | 0.510 | -0.066 | 0.504 | -0.093 |
| No HiC | 0.518 | 0.009 | 0.524 | 0.187 | 0.510 | 0.011 | 0.515 | 0.047 |
| No Overlap | 0.507 | -0.030 | 0.511 | -0.085 | 0.508 | -0.047 | 0.503 | -0.058 |
| No Transformer | 0.861 | 0.484 | 0.846 | 0.353 | 0.843 | 0.413 | 0.825 | 0.391 |
| No Aux Loss | 0.874 | 0.555 | 0.854 | 0.448 | 0.848 | 0.471 | 0.866 | 0.518 |
| No Edge Weight | **0.883** | **0.577** | **0.899** | 0.484 | **0.868** | **0.508** | **0.882** | **0.566** |

Table 5: Performance comparison of different methods across chromosomes. The table shows accuracy (Acc) and Omega Index values for each method on chromosomes 10, 19 (non-acrocentric), 15, and 22 (acrocentric). Bold values indicate the best performance for each chromosome.

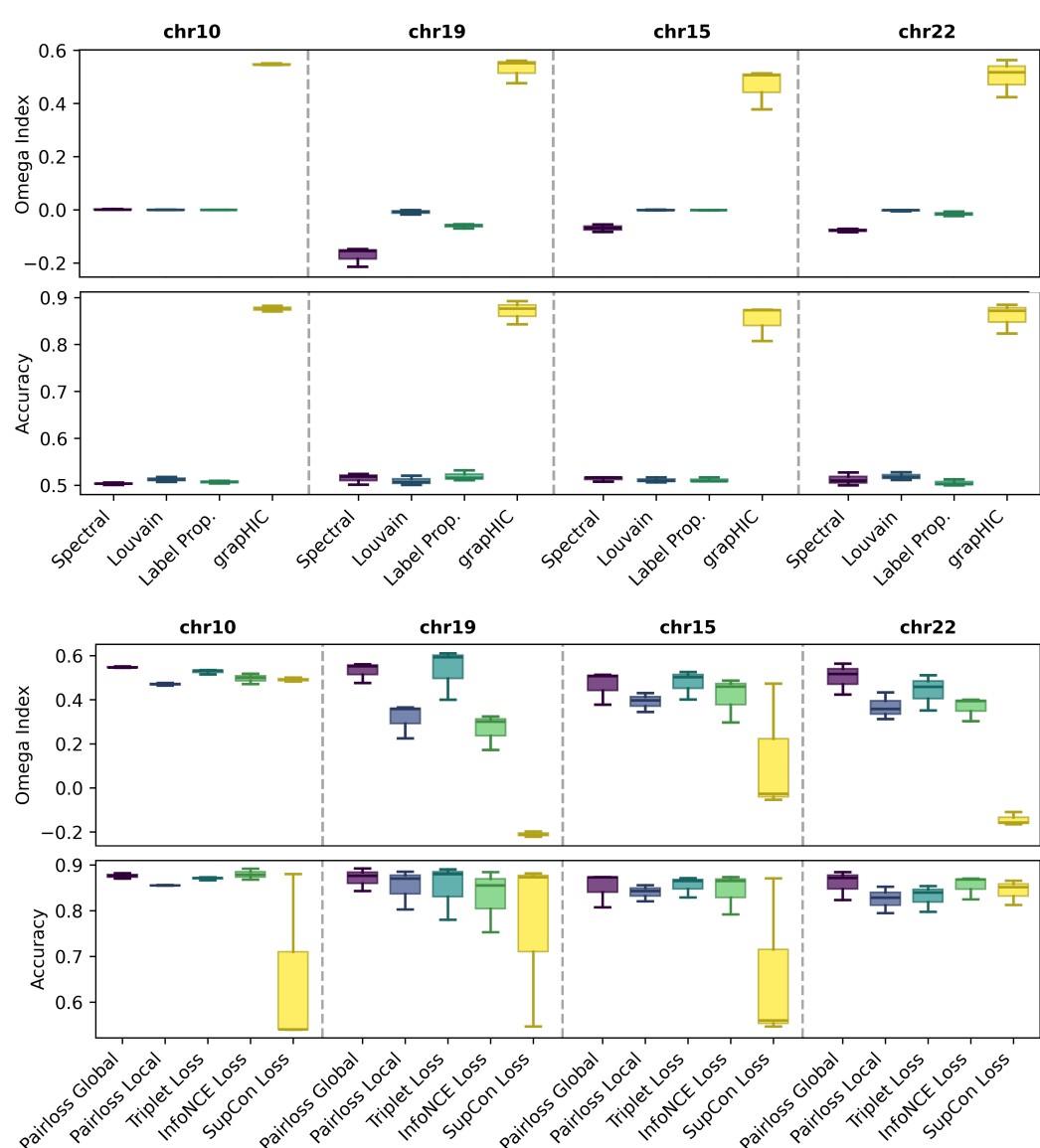

Figure 4: Comparing `grapHiC` with 1) classic clustering algorithms and 2) comparison of differnet contrastive learning functions. Each chromosome is evaluated with three different independently sampled graphs. Note that Chrs. 15 and 22 are acrocentric. The Tukey plot shows the minimum, maximum and median sample as well as standard deviation of the three samples. Accuracy is computed from the best one-to-one mapping as determined by the Hungarian Method between predicted cluster labels and true labels. Omega Index is an extension of Adjusted Random Index (ARI) for fuzzy clustering. The Omega index is an extension of the ARI to fuzzy clustering, and can take values from $-1$ to $+1$, with a perfect clustering being assigned $+1$ and a random assignment $0$. An accuracy of $0.5$ corresponds to random cluster assignments, while $1.0$ would mean perfect clustering.

# I DIPGNNOME + GRAPHIC DETAILS

Figure 6 illustrates the combined approach of `DipGNNome` and `grapHiC` called `D-grapHiC` which we use to obtain the results in our experiments.

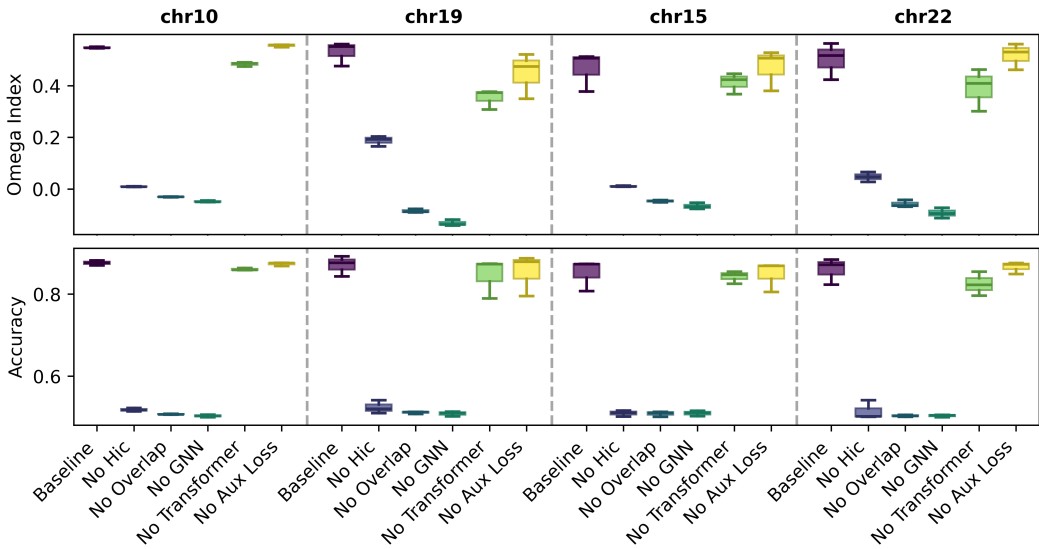

Figure 5: Ablation study for grapHIC. Each chromosome is evaluated with three different graphs sampled from different sets of HiFi reads. The Tukey plot shows the minimum, maximum and median sample as well as the standard deviation of the three samples.

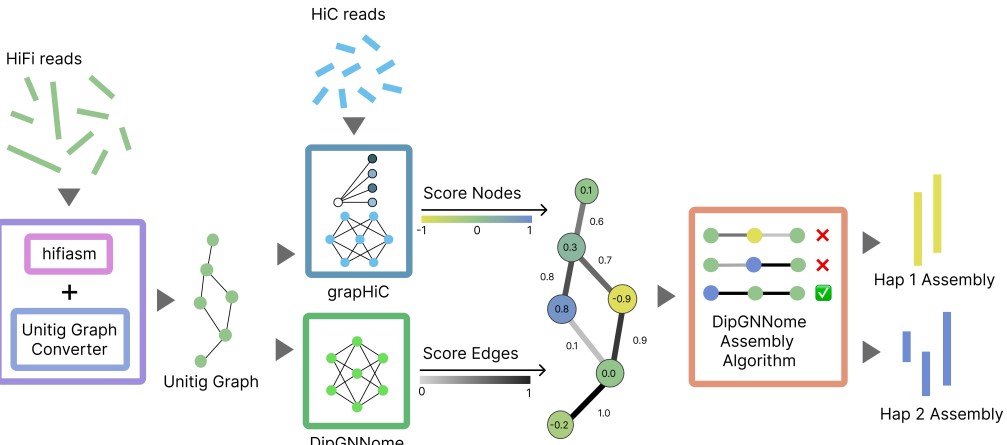

Figure 6: The `D-grapHiC` pipeline begins by constructing a unitig graph from HiFi reads using `hifiasm` together with the `DipGNNome` unitig processor. `DipGNNome`'s GNN then scores the edges of the unitig graph. In parallel, `grapHiC` maps Hi-C reads against the unitigs and assigns node scores representing haplotype probabilities. Finally, a `DipGNNome`-based assembly algorithm integrates the edge scores from `DipGNNome` and the node scores from `grapHiC` to generate a dual-haplotype assembly.

The original beam-guiding heuristic of `DipGNNome`:

$$\mathrm{S}(e) = \alpha \cdot (l(v) - l(e)) - \beta \cdot k_{\neg h}(v) \cdot \frac{l(v) - l(e)}{l(v)} - \gamma \cdot m(e)^2 \qquad (15)$$

where $l(v)$ is the length of node $v$, $l(e)$ is the length of edge $e$, $k_{\neg h}(v)$ is the number of k-mers not matching the target haplotype, and $m(e)$ is the model prediction score for edge $e$.

To replace parental information with `grapHiC`'s HiC-based predictions, we simply replace the non-matching k-mer counts with a penalty based on `grapHiC`'s predicted haplotype.

Without loss of generality, we describe assembling haplotype 1. For this case, we interpret the continuous prediction score $p(v) \in [-1, +1]$ from `grapHiC` as:

- $p(v) > \epsilon$: predicted haplotype 1
- $p(v) < -\epsilon$: predicted haplotype 2
- $|p(v)| \leq \epsilon$: homozygous

We determine the optimal value of $\epsilon$ using the validation set of `grapHiC`. Since the output layer of `grapHiC` uses a $\tanh$ activation, the predictions $p$ lie in the range $[-1, 1]$. For each trained model, we evaluate $\epsilon$ values in the interval $[0, 1]$ with a step size of $0.01$ and select the value that achieves maximal accuracy in distinguishing homozygous from heterozygous nodes.
We then define the following indicator function:

$$\mathbb{I}_{\mathrm{wrong}}(v) = \begin{cases} 1 & \text{if } p(v) < -\epsilon \\ 0 & \text{otherwise} \end{cases}$$

This function assigns a penalty only if the node is confidently predicted to belong to haplotype 2. Homozygous nodes and haplotype 1 predictions are not penalized. (When assembling haplotype 2, we simply negate $p(v)$.)

The updated beam search scoring function becomes:

$$\mathrm{S}(e) = \alpha \cdot (l(v) - l(e)) - \beta \cdot \mathbb{I}_{\mathrm{wrong}}(v) - \gamma \cdot m(e)^2 \qquad (16)$$

This updated heuristic allows the search to favor paths consistent with `grapHiC`'s haplotype predictions in the absence of trio-based k-mer information. The exact beam search parameters used in the experiments of this chapter are given in Suppl. Table 6.

Table 6: Beam search algorithm configuration and heuristic parameters.

| General Parameters | |
| --- | --- |
| Beam width (k) | 5 |
| Samples (n) | 100 |
| Min. contig length ($P_{\min}$) | 100k |
| Min. component length ($C_{\min}$) | 25 |
| **Beam Heuristic Parameters** | |
| **grapHiC + DipGNNome** | |
| $\alpha$ | $10^{-4}$ |
| $\beta$ | 10 |
| $\gamma$ | 20 |
| $c_b$ | $> 0.9$ |
| $c_t$ | $> 0.5$ |

## J  DATA AVAILABILITY

As training data, we used the I002 genome published by Sarashetti *et al.* (Sarashetti et al., 2024a) as the basis for simulation. All PacBio HiFi reads were simulated from the consensus assembly

reported in (Sarashetti et al., 2024a), while real Omni-C reads were used since these are challenging to simulate adequately in eukaryotic genomes across multiple chromosomes. Raw genomic reads are available in the SRA under Accession No. PRJNA1150503. Further details on the I002 dataset can be found in the associated project repository.

For HG002, both reference and read data are publicly available in the HG002 project repository.

## K    CODE AVAILABILITY

`grapHiC` and all scripts used for deployment and plotting in this manuscript are available on GitHub at `https://anonymous.4open.science/r/graphic_iclr-688D/` (repository anonymized for peer review). The repository also contains the best trained model as reported in the results and a link to our training dataset.

