# OpenReview forum: "Binary Node Clustering via Contrastive Learning for Haplotype Phasing in de novo Genome Assembly"
_ICLR.cc/2026/Conference — Submitted to ICLR 2026_

### Official Review · Reviewer_3z9e · 2025-10-31

**Soundness:** 2
**Presentation:** 2
**Contribution:** 2
**Rating:** 2
**Confidence:** 4

**Summary:**

The paper introduces a graph-transformer model, grahpHiC for de novo haplotype phasing in de novo genome assembly. The model assigns values to nodes corresponding to untigs, and subsequent node clustering is used to reconstruct genomes for each haplotype. The graph-transformer model is trained using simulated data with the proposed loss function, supervised contrastive pair (SBP) loss to learn the node values. This is further regularized with an auxiliary loss to improve convergence. Results on experimental data show that global SBP loss usually yields the most accurate phasing; but D-grapHiC usually trails Hifaism in performance when assembling genomes.

**Strengths:**

The key contributions in this work are:

1.	The formulation of the SBP loss function
2.	The graph-transformer model to learn the values/embeddings for each untig node

These are simple, yet possibly useful contributions to the space of phased de novo genome assembly methods. Additionally, the application of graph transformers to the alignment-free setting appears to be a novel contribution.

I would also like to acknowledge the effort made by the authors to provide the reader with some biological background in the introduction and related work sections; this is particularly important for the manuscript to be self-contained and understandable by a reader lacking the requisite background in computational biology.

**Weaknesses:**

There are several clear weaknesses in this paper in its current form. Firstly, the paper initially positions graphic as a de novo genome assembler, but it is effectively only learning untig values to be assembled by some other method (DipGNNome in this paper). This has to be clarified and raises other potential questions, such as whether alternative clustering methods could be effective at assembling genomes after learning the node values.

Moreover, the current landscape of haplotype phasing and de novo genome assembly is somewhat misrepresented. Examples of this are listed below.

1.	Hifaism is not mentioned as a method for phased de novo genome assembly despite being used as a benchmark later.
2.	In the paragraph starting at line 173, the authors state that reference-based haplotype phasing methods have been evaluated on much smaller datasets than is typical of real eukaryotic genomes. However, CAECSeq and XHap (Consul et al, 2023) both demonstrate haplotype assembly on real chromosomal data.

This work also raises a conceptual question – the loss function necessitates the presence of ground-truth labels as it is a supervised loss function. These labels are obtained by virtue of the training data being simulated. With a motivation for the proposed work being that reference-based haplotype assembly methods are biased towards the chosen reference, it is not obvious that the use of simulated data when training grapHiC does not implicitly also introduce a bias towards the reference from which the training data is generated.

Finally, the shortfall of performance of grapHiC vs. Hifaism raises the question of whether the use of machine learning (graph transformer) brings about any tangible benefit in tackling the problem of haplotype-phased de novo genome assembly.

**Questions:**

**Questions:**

1.	ONT reads typically have much higher rates than HiFi reads. How does the performance of the proposed approach vary when ONT reads are used in place of HiFi reads?
2.	What are heterographs (line 106)?
3.	Why is the acronym of “Supervised Contrastive Pair” loss taken to be “SBP”? Should it not be “SCP”?
4.	On line 156, the authors state the this is “the first method capable of phasing raw untig graphs”. As per my understanding, most de novo genome assemblers entail the construction of assembly graphs (k-mers or untigs). How is the described setting different from that of these other works?
5.	Why were other haplotype-phased de novo genome assemblers, such as Hifaism, Falcon-Unzip and HiCanu omitted when describing related work?
6.	Is the random sampling used in the global and local SBP losses fixed across epochs?
7.	What are the parameters used to simulate reads using PBSim3? Also, PBSim3 does not generate HiFi reads; rather, it generates CLR reads that have to be passed to PacBio CSS to generate HiFi reads.

**Suggestions:**

1.	The summary of contributions (lines 155-160) is fairly repetitive as points 2 and 3 are accomplished as part of the first point.
2.	The model architecture in Section 5 would be better described through a pictorial representation. This would improve the readability of the manuscript.

---

> ### Author Response · Authors · 2025-11-18
>
> We thank the reviewer for the in-depth engagement with our work and for highlighting several issues with the current manuscript.
>
> Before answering the reviewer’s questions, we would like to comment on some of the concerns noted in the weaknesses section:
>
> grapHiC is not itself a de novo assembler, but a tool to predict haplotype allocation of unitigs. grapHiCs output can be integrated into a de novo assembler (as demonstrated with DipGNNome in Section 8, replacing parental data with HiC phasing while getting phasing accuracy comparable to hifiasm).
> We were not aware of XHap, and thank the reviewer for mentioning it as related work that supports long reads in Section 3.
>
> Both Xhap and CAECseq report in their publications benchmarking datasets at most 100 kbp (Xhap, in humans) and 5/9.7 kbp (CAECseq, potato/HIV) in size [0]. The Xhap paper itself mentions in section 3.4 and Table 6 running into computational bottlenecks requiring a window-based approach when evaluating Xhap and CAECseq on the smallest human autosomal chromosome, chromosome 22. Other ML-based phasing tools use datasets of similar size, and many only support short reads; in contrast, our focus is on long read data for entire eukaryotic genomes, which are several orders of magnitude larger (~6.2 Gbp in our human datasets, hundreds of Mbp for the chromosome subsamples reported in Fig. 4/5).
>
> Machine learning in genomics is an emerging field, whereas hifiasm has been continuously developed since 2019 (and other classical assemblers have been around since the 1990s).
> We were honestly delighted we were even able to come anywhere close to hifiasm in terms of phasing quality, especially given that new versions of hifiasm generally outperform DipGNNome in most assembly tasks.
>
> It will take time and both conceptual advances and careful software engineering and tweaking for ML-based approaches to match classical genomics in performance. Still, we believe that our contribution of applying a Graph Transformer to phase raw unitig graphs stimulates the field, especially given that phasing on a raw unitig level is something currently not even addressed by classical approaches.
>
> We are also optimistic that working with the hifiasm developers to include support for using our haplotype assignments as input into the hifiasm phasing pipeline could improve performance, in particular for polyploids, where hifiasm’s current heuristics struggle.
>
> Questions:
>
> (1) While ONT reads generally have higher error rates than HiFi reads, read error correction steps like HERRO or the recent version of Dorado can reduce these to levels comparable with PacBio HiFi. In this work, we focused on HiFi reads as these were the state-of-the-art method for assembly of eukaryotic genomes when we started the project, but we plan to explore ONT reads in the future, especially following the recent improvements in Hifiasm 0.25. We have already adjusted our pipeline to work with badreads as an ONT simulator for this purpose, but we focus on HiFi reads as the scope of this first paper.
>
> (2) By heterographs, we mean graphs with different types of edges. We thank the reviewer for pointing out that this is not formally defined in the current manuscript, and we will update Section 2 accordingly.
>
> (3) During the writing process, we changed the name from Supervised Contrastive Pair Loss to Supervised Binary Pair Loss to be more specific to our binary formulation and allow us to make a clear distinction for a future extension to polyploid phasing. We will fix instances where we still use the old name.
>
> (4) All current assemblers known to us perform phasing after assembling contigs from the reads, either in a bubble chain or FASTA representation. This linearized, simplified graph representation is easier to phase, but propagates misjoined contigs. In contrast, grapHiC performs phasing on a raw unitig graph. Our motivation is to leverage a graph transformer architecture that can deal with the messy unitig graph, and allow downstream assemblers to leverage our haplotype assignments and avoid misjoins, particularly for complex genomes
>
> (5) In Section 3, we focus on the machine learning-based phasing literature, after giving some background on genome assembly and briefly mentioning classical Hi-C-based phasers in Section 1. This was meant to keep the paper short and readable for a Machine Learning audience. If preferred by the reviewer, we would be happy to add an additional subsection to Section 3 or expand our discussion of classical approaches in Section 1.
>
> (6) In the current implementation, the random sampling is not fixed across epochs. We could incorporate this by initializing PyTorch’s random seed during validation loss computation. However, we did not experience excessive fluctuations in validation loss.

---

> > ### Comment · Reviewer_3z9e · 2025-11-21
> > **Need to better demonstrate value of grapHiC**
> >
> > I thank the authors for their careful and detailed response. I appreciate the clarification that grapHiC is in itself, not a method for haplotype assembly. However, the discussion of related work is likely to muddle this distinction as it is titled "Deep Learning for Genome Assembly". Given that the reference based methods only seek to reconstruct haplotypes, it would probably be better to distinguish between phasing and assembly here. I acknowledge that haplotype phasing and haplotype phasing are oft used interchangeably but since grapHiC does not directly tackle the problem of assembly, it is important to cast the discussion of related work in a manner relevant to the problem at hand. Additionally, since reference-based method have been shown to be be applicable to larger datasets with adaptations (e.g., sliding window approach), an evaluation of computational complexity (run time) would likelier present a stronger argument for genome assembly using a pipeline with grapHiC + some other tool.
> >
> > In terms of benchmarks, it is noteworthy that the grapHiC-based approach performs comparable to Hifiasm. However, if learning the read phasing comes with additional computational load (training the network for species-specific datasets), then the same argument for the preference of grapHiC-based approaches over reference-based approaches could be used to argue against the utility of this approach. This problem would be exacerbated if it turns out that the model has to be retrained on species-specific or more species-extensive datasets to be utile.
> >
> > Also, the authors have stated that "All current assemblers known to us perform phasing after assembling contigs from the reads, either in a bubble chain or FASTA representation. This linearized, simplified graph representation is easier to phase, but propagates misjoined contigs." This is potentially a good point, but a practical demonstration of the propagaton of misjoined contigs via the simulations/experiments is required to substantiate this argument.
> >
> > Moreover, seeing as (DipGNNome consistently underperforms Hifiasm in existing literature, would it not be more pertinent to set up benchmarks wherein the only difference is the phasing being passed to the genome assembler? For example, in a simulation setting, one could pass the known untig phases as input to the genome assembler. Any other read phasing method could be used to generate the phase values as well. The reason for this suggestion is that the current experimental results capture both the errors introducted by grapHiC and DipGNNome, so it is unclear how well grapHIC is actually performing on experimental data.

---

> > > ### Author Response · Authors · 2025-11-26
> > >
> > > We thank the reviewer for the thoughtful and constructive feedback. We agree that the introduction and related work section should more clearly distinguish between haplotype phasing and genome assembly.
> > > Regarding computational complexity, we appreciate the suggestion. grapHiC performs a forward pass on a comparatively small neural network and takes on the order of tens of seconds to predict phase in the unitig graph of a full human genome.
> > >
> > > Reporting this performance directly would not be a fair comparison for other tools, as the unitigs still need to be joined into phased contigs, a step likely more expensive than our cluster assignment. Aligning HiC reads to the unitig graph is also an expensive step, although this is a highly parallel workload and does not require a GPU.
> > > In our grapHiC + DipGNNome demonstration, the total runtime is dominated by DipGNNome. We will add a paragraph to Section 2 discussing grapHiCs computational performance.
> > > Regarding the training considerations, we agree that demonstrating broader generalization would strengthen the work. Although we expect grapHiC to generalize to assembly graphs of different species like other deep learning tools have demonstrated (e.g. HERRO, GNNome, ), increased diversity would enhance robustness and potentially permit larger networks.
> > >
> > > In our view, this poses a natural direction for future work, also with a view towards polyploid genomes.
> > > Demonstrating misjoined contigs to highlight the advantage of working with the raw unitig graph is a good suggestion, too, and we agree that this would strengthen the manuscript. We should be able to find such cases in our existing DipGNNome vs. hifiasm benchmark by looking for phasing errors occurring inside a single contig. We propose adding a figure demonstrating a case of this happening, along with perhaps some statistics on how often this happens in our dataset (with the expectation that polyploids and more repetitive genomes should have more of these events).
> > > Evaluating grapHiC on simulated data with a known haplotype structure to dissect the impact of different stages in the pipeline is also an excellent suggestion, especially for prioritizing directions for further improvements in a future version of grapHiC.
> > >
> > > Overall, we once again thank the reviewer for their in-depth review and insightful suggestions.
> > > While our current submission already introduces several contributions, such as formulating the diploid unitig phasing problem as a graph-based clustering task and finding a suitable loss function respecting its inherent symmetries, training a graph transformer using this loss, and implementing an end-to-end phased assembly pipeline using grapHiC and DipGNNome, the reviewer’s suggestion strengthens both this manuscript and future work on grapHiC.

---

> > > > ### Comment · Reviewer_3z9e · 2025-11-27
> > > >
> > > > I am happy to see the author-reviewer engagement and appreciate the effort made by the authors to address the reviewer concerns. However, I think the paper is still clearly lacking in terms of representative benchmarking to make a convincing case for the proposed algorithm. Addressing this will require a fairly extensive revision of the manuscript, hence I am maintaining my original score.
> > > >
> > > > That being said, I am intrigued by the presented work and look forward to seeing the published version. A cross-species deep learning untig graph construction (as returned by grapHiC) could have some very interesting applications, even beyond genome assembly.

---

> ### Author Response · Authors · 2025-11-18
>
> (7) We used PBSim’s distribution matching feature (--method sample) to match the error and length distribution to a set of HiFi reads aligned to the HG002 reference. We apologize for being imprecise in our description of the read simulation, and will update Appendix D accordingly.
>
> We again thank the reviewer for their detailed review and clear suggestions for improving the manuscript, as well as for mentioning Xhap – we were not aware of this work previously. We hope that our answer resolved many of the reviewers’ concerns, and welcome additional suggestions.
>
> [0]
> CAEC-Seq: “The reference genome 5000 bp long is randomly selected from Solanum Tuberosum [sic] chromosome 5” from Section 3.1, Page 6 in Ziqi Ke and Haris Vikalo, NEURIPS 2020. In Section 3.2 on the following page a HIV full genome dataset is introduced, but the HIV genome is only approx. 9.7 kbp, far off any eukaryotic or even bacterial genome.
> Xhap: “[...] we select at random a 10 kb region of the Solanum tuberosum
> Chromosome 5 as the reference [...]“ in Section 3.1 Page 6 from Shorya Consul, Ziqi Ke, Haris Vikalo, Bioinformatics Advances, 2023, 00, vbad169.
> In Section 3.2 on page 8, they introduce a longer dataset for long-read experiments (“[...] we first select at random a 100 kb region of the human GrCh38 genome [...]”).
> In 3.3 on the same page, a third dataset consisting of 5 10 kbp regions from S. tuberosum genome is introduced.
> In section 3.4 on the following page, an analysis of the entire human chromosome 22 is reported, though that employed a window-based approach:
> “Owing to the large size of the read fragment matrix, we run XHap to reconstruct overlapping haplotype blocks and phase them together to obtain the complete reconstructed haplotypes.”
> As can be seen in Table 6, this required substantial compute (~50h for XHap, ~3580h for CAECSeq), while other tools ran by the authors crashed on this larger dataset.
> Note that CAECSeq and XHap both cite Motazedi et al. in Brief Bioinform. (2018) 1;19(3) for their pipeline.

---

### Official Review · Reviewer_CEqa · 2025-11-01

**Soundness:** 3
**Presentation:** 3
**Contribution:** 3
**Rating:** 4
**Confidence:** 4

**Summary:**

This paper proposes a graph-based deep learning framework for haplotype phasing in de novo genome assembly. The method, named grapHiC, operates on assembly or unitig graphs, where nodes represent sequence fragments and edges encode sequence overlap.
The main idea is to cast phasing as a binary node clustering problem and train a contrastive graph neural network that separates nodes belonging to different haplotypes while preserving local contiguity. Experimental results on both simulated and real sequencing datasets show that grapHiC achieves high N50 and low switch error rates.

**Strengths:**

The paper addresses an important and technically challenging bioinformatics problem, reference-free haplotype phasing, using modern graph representation learning. The presentation is clear and well-organized, making the ideas accessible to both ML and bioinformatics audiences. The binary node clustering formulation is intuitive and makes a clear connection between genome assembly graphs and graph learning objectives.

**Weaknesses:**

The conceptual novelty is somewhat limited: while grapHiC differs from prior graph-based phasing methods such as GAEseq and NeurHap by operating in a reference-free de novo setting, the underlying modeling approach remains similar.

No deeper theoretical grounding or interpretability of the framework is offered.

The experimental evaluation is limited in scope. Table 1 compares grapHiC to generic graph-clustering algorithms (Spectral, Louvain, etc.), which are not meaningful phasing baselines; this shows improvement over standard community detection rather than true domain-specific state-of-the-art methods. Table 2 is also ambiguous: if I understand its use here correctly, hifiasm is used to generate the assembly graphs and to produce the reference phasing statistics, so the reported "comparable performance" might simply mean that grapHiC can reproduce hifiasm’s own phasing on those same graphs (i.e., not that it matches hifiasm as an independent de novo phasing method).

**Questions:**

The paper should include comparisons to domain-relevant de novo phasing tools (e.g., DipGNNome and a few others) to better establish novelty.

The role of hifiasm in the pipeline should be explained in more details (e.g., is it a pre-processing step or a competing phasing baseline).

It would be helpful if there is a deeper discussion about how the proposed loss fundamentally differs from standard contrastive or cross-entropy losses used in graph partitioning.

How much does performance depend on the Hi-C linkage density or noise level?

Scalability metrics (runtime, memory) and sensitivity to graph size or read coverage should be reported.

A discussion of potential generalization of the approach to polyploid or metagenomic assembly graphs would be beneficial.

---

> ### Author Response · Authors · 2025-11-18
>
> We thank the reviewer for their comments and their appreciation of our attempt to make the manuscript read well to both ML and bioinformatics-oriented audiences.
> We address the questions raised in order:
>
> (1) We have worked closely with the developers of DipGNNome for the grapHiC integration reported in this manuscript, and are looking to continue with regard to future directions (see answer to question 6). As briefly outlined in Section 8 and the introduction, we think the main advantage of using grapHiC when running DipGNNome is replacing parental sequencing, requiring additional sample collection and consent (for human samples), with a far easier HiC step.
>
> We’re also in contact with the developers of hifiasm and plan to incorporate grapHiC’s assignments of unitigs to haplotypes as an input to hifiasm's internal heuristics.
> Regarding GAEseq, NeurHap, Ralphi, and other GNN-based phasing tools, grapHiC solves a somewhat different problem both in terms of input data and problem formulation, as briefly outlined in Section 3.1. This ML phasing literature works on far simpler reference-based read-overlap graphs obtained from a SNP matrix.
>
> Making grapHiC’s output useful for phased de novo genome assembly tools required dealing with messy, large raw unitig graphs and incorporating HiC information, which we accomplished by using a bespoke loss and graph transformer architecture.
> To summarize our contribution, we see grapHiC as closing a gap between the ML phasing and genome assembly worlds. If this does not come through clearly from the current manuscript, we are happy to expand the conclusion to discuss this more explicitly.
>
> (2) hifiasm’s role in the pipeline is limited to generating the raw overlap graphs from HiFi reads, as briefly explained in Section 6.1 and shown in more detail in Figure 3 of the Appendix. Our pipeline performs all subsequent graph simplification and phasing steps. The predicted phase of each unitig is provided as input to DipGNNome to obtain a phased assembly, and this is compared against hifiasm’s assembly output in Table 2. Because both methods start from the same raw hifiasm graphs, our graph simplification and phasing capabilities can be compared to those of hifiasm.
>
> To summarize briefly, we use hifiasm for its highly efficient read overlap detection, and grapHiC and DipGNNome for all downstream steps (acting as a competitor to hifiasm). We are happy to consider making this clearer in section 6.1 and consider moving Fig. 3 to the main text.
>
> (3) The proposed loss differs from standard contrastive learning losses because it is specifically adapted to a 1-dimensional embedding space for the binary clustering problem of diploid phasing. It is also symmetric with regard to swapping the haplotype labels per chromosome, as these are arbitrary in the absence of parental data. This represents the special features of the phasing problem better than a generic loss, as we demonstrate in Table 1, where we compare performance to training the same network on an InfoNCE and SupCon loss.
>
> (4) In Appendix J, we report an ablation study of different components of the grapHiC model to investigate factors affecting its performance.
> For every training run, HiC reads were randomly downsampled to 15 Gbp. We did not intentionally perform artificial noising of our HiC data or investigate the performance of downsampling to different coverages. However, to facilitate rapid prototyping, we developed a faster version of the HiC processing pipeline using minimap2 instead of bowtie2 and a faster normalization approach (described in more detail in Section 6.1). This faster pipeline should have more noise present in the HiC signal, as minimap2’s minimizer approach is not ideally suited to aligning short reads. Training the model on the faster pipeline did not seem to negatively affect model performance.
>
> (5) Details on computational resources used are provided in Appendix F.
> In Figures and Tables 4 and 5 in the Appendix, we evaluate grapHiC on different human chromosomes of different sizes, ranging from 135 Mbp (Chr10) to 50 Mbp (Chr 22). Two of the compared chromosomes are acrocentric, which tend to have more difficult assembly graphs due to their high repeat content.

---

> ### Author Response · Authors · 2025-11-18
>
> (6) As very briefly mentioned in the Conclusion due to space constraints, we believe extending the grapHiC approach to polyploids is a promising avenue, as classical heuristic methods struggle in these more complex settings. This was also our motivation for working with more complex overlap graphs instead of on a contig level, as contig misjoins are more of a concern the more haplotypes are present in the sample. Extending grapHiC will require generalizing our loss function to multidimensional embedding spaces while maintaining the symmetries we are exploiting for improved performance. We’re currently exploring some approaches in this space, working on a recently published dataset of high-quality autotetraploid potato assemblies (Sun, et al. Nature 642, 2025).
>
> Phasing in metagenomics is another interesting avenue for extending grapHiC. Challenges here include the unknown number of strains present in a sample, as well as the scarcity of high-quality ground truth datasets to train and evaluate on. Perhaps using a CAMI dataset would allow tackling this challenge.
>
> We thank the reviewer for raising this point, as we are also excited about these follow-up works for grapHiC.

---

### Official Review · Reviewer_T4Ve · 2025-11-01

**Soundness:** 3
**Presentation:** 2
**Contribution:** 3
**Rating:** 6
**Confidence:** 3

**Summary:**

The authors introduce a new algorithm for genome phasing, i.e., splitting a genome into haplotypes. The key advantage of their approach is that it does not rely on a reference sequence. Instead, it leverages a Hi-C graph using a graph transformer to perform the phasing based on the idea of an enrichment of contacts on the same haplotype . The method is applied to the unitig graph generated by the hifiasm assembler.

**Strengths:**

1. High novelty (use of Hi-C + graph transformer + bespoke loss functions), and great application of machine learning to the key problem of phasing in genomics (without a reference genome)
2. Strong results in benchmarking
3. Good ablations and baselines against traditional algorithmic approaches for clustering

**Weaknesses:**

1. There is a degree of circularity in the approach. On the one hand, the method is intended for use in settings where no reference genome is available; on the other hand, it is trained and benchmarked using labels derived from reference genomes. While the results are still strong, I would have liked to see an example where the model is trained on one species and then applied to reads and Hi-C data from a different species.
2. I might have missed it, but the main body of the manuscript doesn't include in the benchmarking reference-only phasing baselines

**Questions:**

1. Can the authors attempt to produce an assembly for a different species using the model trained on human data?
2. Please add a baseline from phasing using only a reference genome

---

> ### Author Response · Authors · 2025-11-18
>
> We thank Reviewer 1 for the comments and their appreciation about the novelty of our approach.
> Below, we address the two questions raised in the review:
>
> 1. We agree that cross-species generalization is an important point. In future work, we plan to generalize our losses to polyploid genomes and deploy grapHiC on a wider variety of species.
> For now, we think the results of 6 samples in two different high-quality human genomes are an encouraging proof of concept for grapHiC’s approach.
>
> 2. Existing ML-based phasing tools, such as NeurHap, GAEseq, CAECSeq, and ralphi, were trained and evaluated only on small chromosome segments or on small bacterial/viral data sets.
> To the best of our knowledge, no ML-based tool has been developed that scales to long reads of whole eukaryotic genomes.
> Classical approaches can scale to these datasets, but phase on a contig or scaffold level (e.g., hifiasm, Greenhill, HapHiC). This simplifies the problem, but also means they cannot recover misjoined contigs.
>
> Regarding experimental design, our considerations were to be the first to consider the phasing problem in a node-clustering framework, in which the natural basis for comparison is classical clustering approaches. As a proof of concept, we incorporate our method into the DipGNNome assembler and compare phasing metrics to the current state-of-the-art OLC assembler, hifiasm. Making grapHiC compatible with hifiasm could be a promising future step towards making grapHiC useful to the genome assembly community.

---

> > ### Comment · Reviewer_T4Ve · 2025-11-26
> >
> > I appreciate the authors’ response and will maintain my score. I am looking forward to the further developments they have outlined.

---

### Meta-Review · Area_Chair_3w4R · 2025-12-22

**Summary:**

- Most reviewers raised concerns about the experimental evaluation and the performance relative to SotA methods on benchmarks
- Most reviewers also raised concerns about the conceptual novelty / missing  theoretical grounding.
- Some reviewers raised questions about unclear cross-species generalization.
- There was also some discussion about  the description of grapHiC as a de-novo genome assembler (which is not correct).
- There were also concerns about the use of simulated data and potential bias issues.

**Reviewer Concerns:**

I think, the concern regarding the proper  description of grapHiC could be addressed in the rebuttal, but this discussion led to the more fundamental concern that in the whole paper, a clearer distinction between phasing and assembly would be needed, for which a substantial revision might be necessary.

In my opinion, the concerns about the benchmark experiments could not be addressed in a fully convincing way, and also some questions about the methodological novelty/depths are still open.

**Reviewer Scores:**

I don't think that any review scores changed significantly after the rebuttal.

---

### Decision · Program_Chairs · 2026-01-26

Reject